# Assessment and Prediction of Fish Freshness Using Mathematical Modelling: A Review

**DOI:** 10.3390/foods11152312

**Published:** 2022-08-02

**Authors:** Míriam R. García, Jose Antonio Ferez-Rubio, Carlos Vilas

**Affiliations:** 1Research Group on Biosystems and Bioprocess Engineering (Bio2eng), IIM-CSIC, 36208 Vigo, Spain; miriamr@iim.csic.es (M.R.G.); jaferez@cebas.csic.es (J.A.F.-R.); 2Research Group on Microbiology and Quality of Fruit and Vegetables, CEBAS-CSIC, 30100 Murcia, Spain

**Keywords:** mathematical modelling, fish quality, fish freshness, bibliometric analysis, predictive microbiology, stress variables, quality degradation

## Abstract

Fish freshness can be considered as the combination of different nutritional and organoleptic attributes that rapidly deteriorate after fish capture, i.e., during processing (cutting, gutting, packaging), storage, transport, distribution, and retail. The rate at which this degradation occurs is affected by several stress variables such as temperature, water activity, or pH, among others. The food industry is aware that fish freshness is a key feature influencing consumers’ willingness to pay for the product. Therefore, tools that allow rapid and reliable assessment and prediction of the attributes related to freshness are gaining relevance. The main objective of this work is to provide a comprehensive review of the mathematical models used to describe and predict the changes in the key quality indicators in fresh fish and shellfish during storage. The work also briefly describes such indicators, discusses the most relevant stress factors affecting the quality of fresh fish, and presents a bibliometric analysis of the results obtained from a systematic literature search on the subject.

## 1. Introduction

The main causes of food discarding among consumers and retailers are the food aspect, outdating, and safety uncertainty [1]. Damage and spoilage of foods lead to around 15% of waste, which increases to 35% if food is subject to inadequate storage and transport conditions [2,3]. Mathematical modelling describing the evolution of food quality indicators, under given storage and transport conditions, is central to minimising food waste [2]. Therefore, the prediction of fresh fish quality is a major challenge for the food industry, distributors and retailers to adjust prices and minimise food waste.

Fresh fish and shellfish are highly perishable products due to their biological composition. Under normal handling chilled or refrigerated storage conditions, their shelf life is limited by enzymatic, chemical and microbiological spoilage. Fresh fish is stored, transported and distributed in boxes of high-density poly-ethylene filled with ice. Other common conservation methods for fresh fish are the transport and storage in tubs with water and ice or at superchilling temperatures [4,5]. From fish capture to consumer consumption, there are several factors affecting fish quality, being temperature the most relevant. The prediction of fish quality in all these cases is critical to determine the price of the product and sell it before it is not of sufficient quality, or even safety, for the consumer.

Quality in fresh fish, and generally in food, is a broad concept that involves different attributes (chemical, physical, microbiological, sensory) which can be measured either directly or indirectly. In the last decades, several analytical techniques have arisen [6] including biosensors to measure microbial pathogens or biogenic amines [7]; electronic noses or electronic tongues for volatile compounds, K-value or TVB-N [8,9]; or hiperespectral imaging to determine moisture content or texture [10,11]; among other. The selection of the analytical method depends on the selected indicator, but also on properties such as accuracy, reliability, portability, rapidity, easiness to use and analyse the results, time consumption and price. Ideally, the methods should be also non-destructive and non-invasive [12]. Recent works [6,13,14] present exhaustive reviews regarding the different analytical methods considered in the literature to measure the most commonly used quality indicators for fish freshness assessment. Typically, assessment methods focus on fish quality at the moment of measurement but are unable to predict quality changes in the following days. Prediction requires the use of appropriate mathematical models. It is important to mention that some authors, for instance, ref. [15,16,17], use the term *predictive model* to denote models that correlate freshness indicators with experimental measurements (pH, TVB-N, hyperspectral imaging, Electronic nose data, etc.). Typically, some type of regression is used to obtain these models. Although these works are common and necessary, in this review, we will use the term *predictive* for those models able to forecast the future evolution of the freshness indicators. Otherwise, we will use the term *assessment* or *estimation*. Mathematical models for the prediction of fresh fish quality are diverse and difficult to classify.

In this review, we propose the use of general features of the mathematical structure to organise the different modelling alternatives, as illustrated in Figure 1. The final objective of the model is either to estimate (using indirect online measurements) or to predict one or more chemical, physical, microbiological or sensory attributes that are indicators of the consumer’s perception. Such attributes (odour, texture, TVB-N, spoilage bacteria, etc.) are the output of the model (blue part of the figure). The input of the model (red part of the figure) can be any external factor (temperature, pH, fish feed, etc.) affecting the quality indicators. The model (orange part of the figure) provides a quantitative relationship between inputs and outputs either using empirical relationships or inspired by known mechanisms of fish quality degradation.

This review is organised following the structure presented in Figure 1. It begins by summarising the results of the systematic literature review performed to investigate the most typical quality attributes used to define fish quality, including a brief description of the procedure followed to perform such a review. Secondly, it introduces the relevant stress factors affecting fish freshness. Such stress factors can be split into two groups: Pre-slaughter/slaughter conditions, and handling, storage and distribution conditions. In Section 4, we review the mathematical relationships or models that allow us to describe and predict fish quality attributes (outputs) as a function of the stress factors (inputs). As a final remark, we will discuss what we think are the main challenges for modelling fish quality and possible alternative solutions to consider in the future. As a general rule, we will denote by fish quality any positive attribute, either nutritional, organoleptic or a combination, in fresh fish, and therefore related to fish freshness. In this regard, issues not related to fresh fish were not considered in this review, such as processed fish (cooked, sterilised, etc.) or social aspects such as food security or production of added-value compounds.

## 2. Quality Attributes (Model Outputs)

Food quality in fresh fish and fresh fishery products is a broad and complex concept defined by a set of attributes (quality attributes) that are either nutritional, organoleptic, or a combination of both. The levels (concentrations) of some chemical compounds describe nutritional quality attributes. Examples of nutrients include vitamins, bioactive forms of oligo-elements, essential amino acids, digestible proteins or unsaturated fatty acids, among others. On the other hand, colour, texture, flavours or aroma are attributes defining the organoleptic quality of a particular food product. Organoleptic variations in fish are caused by changes in chemical, microbiological and physical properties. Sometimes, quality attributes are defined by the level of a particular biological, chemical or biochemical factor (e.g., the concentration of a given vitamin, nucleotide, enzyme, or bacteria, among others), although usually, it is the result of a combination of different factors. For example, colour is the (observable) result of a certain combination of pigments on a given food matrix that have been produced or consumed under the action of many biochemical transformations. Other quality attributes, such as freshness, are defined by the combination of nutritional and organoleptic properties which deteriorate with time. Freshness can quantitatively be described using different sensory scores, such as the Quality Index Method (QIM), or other simpler indicators, such as the shelf life dating of a given food product, either to specify the last date when the product must be sold (“sell by date”), the “high-quality” period or the date when the product must be removed from the store [18].

We have analysed this diversity of quality attributes studied in the literature, and their interconnections, by conducting a systematic bibliographic review and bibliometric analysis of the articles (in the *Web of Science Core Collection* database, *Food Science and Technology* category). Figure 2 summarises the procedure we have followed to perform the systematic literature search.

Inside the dashed rectangle, we represent the iterative part of the procedure. First, we constructed two lists. One of the lists contained terms related to fish quality, such as freshness, K-value, and lipid oxidation, among others. The other one contained general terms such as seafood or shellfish and fish species names. In the search, words from both lists must appear in a major field (abstract, title or author keywords). We have found many works that included words from both lists, in which the quality terms were not related to fish. To avoid these situations, we used a maximum separation of five words (NEAR/5) between the terms in both lists. This separation was chosen by the trial and error method with the objectives of avoiding the exclusion of manuscripts within the scope of this work (shorter separations) and minimising the number of off-topic manuscripts (larger separations). Since the focus of this review is on fresh fish, we have excluded from the search (NOT) those words related to off-topic issues such as processing (sterilisation, modified atmosphere, etc.), production of added-value compounds (essential oils, gelatin, etc.), food security or other social aspects, among others. The initial search still included works not related to fresh fish quality so we identified, new terms that should be avoided, and repeated the search. The list of terms included and excluded in the systematic search is presented in Appendix A. This iterative procedure resulted in 1636 works. Details about such works and a bibliometric analysis are available online in the repository [19]. In this material, the interested reader can access 2 types of reports: a PDF file that includes the main results of the general search, and 12 interactive reports for sub-collections defined as the group of works where one of the 12 selected quality attributes is mentioned in a major field. Table 1 summarises the results of the systematic review for each of the 12 categories or groups. The table presents, for each category, the number of works, total citations, average citations per work, and the most cited works.

Additionally, we have incorporated in the repository a PDF file that contains the information included in the interactive reports. All the reports, iterative or not, include a general analysis of the collection (including, for example, time span, collaboration index, authors per article, citations per year), and analysis of the countries, authors, articles and journals. Networks of co-citations and keyword co-occurrences were also incorporated. Moreover, the collections can be downloaded as an Excel file and the graphs and tables can be manipulated to, for example, constrain the works to those where certain words, selected by the user, are mentioned in the abstract.

Most of the manuscripts were research articles, dating back from 1949 and with citations increasing homogeneously since then, except for two jumps in publications in 1977 and 2002. Around 100 works were published in 2021, the last year considered in the review. The most cited works were written by Ólafsdóttir et al. [22] with a review on methods to evaluate fish freshness and by Ryder [46] with a method to measure ATP and its breakdown products to estimate the $K_{I}$-value. The most productive countries have been China, USA and Spain. These are also the countries where the documents with the greatest impact (in terms of average article citations) were produced. Journal of Food Science (145 documents), Food Chemistry (140 documents) and Journal of Agricultural and Food Chemistry (72 documents) are the journals with the largest number of publications. The most productive authors were from the northwest of Spain (S. P. Aubourg, Barros-Velázquez), whereas the authors with the greatest impact were Ólafsdóttir and Dalgaard from Iceland and Denmark, respectively.

After the iterative procedure was finished, we refined the search (see Figure 2) to find those works related to mathematical modelling. We read the resulting 33 manuscripts and kept those (25) that fitted within the scope of this review.

In the next sections, we define each of the attributes with references to the main works in the area. For a deeper discussion about these quality attributes we recommend some recent comprehensive reviews [6,13,14].

### 2.1. Lipid Oxidation

Lipid oxidation is the attribute gathering the greatest attention, both in terms of the number of citation counts and of works, with a remarkable increase of publications during the first decade of the current century remaining almost invariant during recent years. The most cited researchers on this topic are M. P. Richards and Y. Ozogul. Although fatty acids are also nutrients (described in Section 2.10), we have decided to consider them separately because of the importance of lipid oxidation. Fish constitutes the main source of polyunsaturated fatty acids (PUFA), up to 40% of long chain fatty acids [47], with quantity and composition changing with the catching period and depending on whether the fish is wild or cultured [21]. PUFA are easily oxidised into aldehydes, responsible for changes in flavour, texture and odour, known as rancidity [6,12,20,22,48], and are highly affected in most of the cases by previous fish bleeding [20]. In addition, the oxidation reaction can decrease the nutritional quality of food and certain oxidative products are potentially toxic [48]. Primary lipid peroxidation products (peroxide value being its most common measure) include hydroperoxide that is unstable and decomposes to generate various secondary products, such as aldehydes that contribute to fish rancidity [12]. The most common method to measure aldehydes is the thiobarbituric acid-reactive substance (TBARS) test [22].

The main advantage of this indicator is that the oxidation of unsaturated lipids produces alterations in smell, taste, texture, colour, and nutritional value [22]. Therefore, it provides us with a global measure of fish freshness. However, analysis of PUFA is a destructive method so the fish sample analysed cannot be commercialised.

### 2.2. Sensory Analysis

Despite being the second most relevant quality attribute in terms of publications (with G. Ólafsdóttir being to a large extent the most cited author), sensory analysis is the most used method to assess freshness, probably because it depends on a combination of the other quality attributes [22]. In this group, we include any form of measure or interpretation of fish freshness perceived by the senses of sight, smell, taste, or touch. They can be assessed by a trained panel or by consumers’ subjective opinions about preferences. The most common method in Europe is the Quality Sensory Method (QSM) based on the Council Regulation (EC) No 2406/96 for marketing standards [49]. The output of this method is a discrete value that classifies fish in four levels of freshness (Extra, A, B, and Not Admitted). This method is based on shared characteristics in fresh fish and therefore is common to different fish species. For specific tests, the most common method is the Quality Index Method (QIM) [50]. The output of the QIM is also a discrete value (0,1,2,…,n), where lower values correspond to fresher fish. The criteria to select the value and the number of levels (*n*) depend on the fish species considered.

The main advantages of these methods are: (i) they are minimally invasive, and do not involve the destruction of the sample, since the sense of taste is not included; (ii) they can be used to estimate fish shelf life by agreeing from which level the food is not considered of sufficient quality to be sold to the consumer; (iii) instead of focusing on one particular feature, they provide a global evaluation of fresh fish. However, sensory analysis methods are highly subjective and depend, to a large extent, on the expertise of evaluators. The cost associated with the use of a panel of evaluators is another disadvantage. These disadvantages can be alleviated by using analytical techniques -colourimeters, electronic noses, hardness testers- instead of a panel of experts.

Another alternative is the Global Stability Index (GSI), a score gathering the influence of many attributes, sensory or not. The GSI is computed using the following general expression [51]:(1)$$GSI=1-\sum_{i=1}^{n}\alpha_{i}\frac{A_{i}-A_{i,0}}{L_{i}-A_{i,0}}$$

$A_{i}$ and $A_{i,0}$ are, respectively, the values of a given attribute, for example, TVB-N or K-value, at assessment time *t* and at initial time t=0. $\alpha_{i}$ is the weight given to each attribute, *n* is the total number of attributes considered, and $L_{i}$ is the spoilage threshold for attribute $A_{i}$.

### 2.3. TVB-N/TMA-N

The formation of volatile nitrogen bases, such as trimethylamine (TMA-N), dimethylamine (DMA), or ammonia, from the reduction of trimethylamine oxide (TMAO), is a widely investigated cause of fish odour. There was a substantial increase in interest during the 10 years after 1996, being Ruiz-Capillas the most cited author. The total amount of volatile bases (TVB-N), as well as individual methylamines, have been extensively used as indicators of quality degradation in postmortem fish [52,53]. The reduction reactions are catalysed by bacteria such as *Shewanella* spp. or *Pseusomonas* spp., during fish spoilage [54]. Some authors [17,26,52] have argued that these volatile bases are poor freshness indicators for some fish species because of the low content, or even absence, of TMAO. However, the FAO specified a maximum allowable level in international trading of 10 mg of TMA-nitrogen per 100 g fish muscle [55].

### 2.4. Spoilage Bacteria

Research works regarding this quality attribute have increased since 1994 without any deceleration in the last years as previously described attributes. Dalgaard, Gram and Bulushi, are, markedly, the most cited authors. Fish freshness deteriorates rapidly with the growth of Gram negative psychrophilic or psychrotrophic bacteria, named spoilage bacteria (SSO), due to their ability to reduce TMAO and to produce hydrogen sulphide [28]. Common spoilage bacteria of fish at chilled temperatures are: (1) *Shewanella putrefaciens* for being $H_{2}S$-producing bacteria and with acceptability limits around $10^{7}$ CFU/g [56] or even slightly higher $10^{7.02}$ CFU/g [57], (2) *Pseudomonas* spp. with the same acceptability limits [56,57,58], or larger when analysing for example *Pseudomonas psychrofila* ($10^{8.5}$ CFU/g [59]), and (3) broad measurements such as Total Viable Counts (TVC) with a limit of $10^{6}$ CFU/g [16]. More information can be found in [28,60,61].

Since microbial growth and metabolism is the major cause of food spoilage [62], spoilage bacteria is a key indicator of fish freshness and shelf life. The main disadvantage is that the procedure to determine bacterial concentration is tedious and time consuming.

### 2.5. Texture Properties

Bibliometric analysis for texture attributes reveals that many works mentioned this attribute without being the focus of the work. Among the works considering texture as a relevant focus, we would like to highlight the review on fish texture [63] and a research article assessing texture, among other chemical and sensory characteristics, of sea bream [30].

Texture can be evaluated using minimally invasive techniques. However, contrary to other food matrices, such as beef, texture in fish is not usually regarded as a relevant freshness indicator, particularly when considering fresh fish. As mentioned in [64], it should be considered in combination with other indicators, such as colour and odour. Use of texture as a fish quality indicator limits to either cooked or frozen fish stored for long periods, where an increase in toughness and dryness of the tissues can be observed [63]. After cooking, the taste of fresh fish is associated with firm meat that goes to dry, crumbly with short, tough fibres for deteriorated fish. However, raw fish maximum toughness commonly occurs after 1–2 days of storage, corresponding to the minimum pH and *rigor mortis* [63]. Texture properties are typically considered in the evaluation of the QIM [65].

### 2.6. ATP Degradation

Although the number of citations places this attribute in the sixth position, it is the third one in average citations per work. Most of the works were published in the nineties and during the past 10 years, with G. Ólafsdóttir being the most cited author.

After fish death, ATP transforms, within the first 24–48 h, into inosine 5’-monophosphate (IMP) in three steps, producing adenosine diphosphate (ADP) and adenosine monophosphate (AMP) [66,67]. IMP degradation continues on a cascade of reactions that produces Inosine 5${}^{\prime }$-monophosphate (Ino) and hypoxanthine (Hx), which is further decomposed in other compounds such as xanthine and uric acid [52,67]. This cascade of reactions occurs in the order of days to weeks, depending on the storage temperature and bacterial concentration [68]. IMP is related to the pleasant sweet and meaty flavors in fresh fish, *umami* flavour [69,70], whereas Hx is responsible of unpleasant bitterness [71,72]. The K-value and K${}_{I}$-value are two of the most widely employed indicators to evaluate freshness. They are defined as the following function of the ATP degradation products [68,73,74,75,76]:(2)$$\mathrm{K-}\mathrm{value}=\frac{\mathrm{Ino}+\mathrm{Hx}}{\mathrm{ATP}+\mathrm{ADP}+\mathrm{AMP}+\mathrm{IMP}+\mathrm{Ino}+\mathrm{Hx}}\;\cdot \;100$$
(3)$$K_{I}\mathrm{-v}alue=\frac{Ino+Hx}{IMP+Ino+Hx}\;\cdot \;100$$

The K-value has been also correlated with the freshness of different fish species [73]. The authors found that *very high* grade corresponded with K-value lower than 10%. *High* grade individuals had K-values lower than 20% or 30%, depending on the fish species. Fishes with K-value up to 50% correlated with *medium* grade. Finally, K-value larger than 50–70% was obtained for *low* grade samples.

The main advantages of indexes based on the degradation of ATP (K-value, $K_{I}\mathrm{-v}alue$) are their reliability [66] and, as mentioned above, their direct connection with fish flavor. The main disadvantages are that the evaluation of K-value or $K_{I}\mathrm{-v}alue$ requires the destruction of the sample and their usefulness depends on the fish species being examined [64].

### 2.7. Biogenic Amines

Works studying the correlation between spoilage and biogenic amines are homogeneously increasing since the nineties, being the most cited articles by Bulushi, Ruiz-Capillas and Vecianogues. Biogenic amines here are non-volatile amines (histamine, cadaverine and putrescine) formed by decarboxylation of amino acids (histidine, lysine and ornithine, respectively). Although TMA-N and TMAO are also biogenic amines, they are not considered in this group because TMA-N is volatile and it is a result of the degradation of TMAO, therefore being one of the main contributors to the formation of TVB-N, as discussed in Section 2.3. Within non-volatile biogenic amines, histamine is the most studied due to its toxicity and allergic potential, but it is unable to correlate with the level of spoilage for different fish and conditions [23]. Cadaverine is the biogenic amine that can be used as a spoilage indicator (for example, for salmonid fish, values less than 10 mg/kg indicate good quality [77]). Putrescine, however, is not a good spoilage index because its amine, ornithine, is not present in all fish species (for example, it is missing in tuna). Alternatively, there are amine indexes combining different biogenic amines such as the amine index (AI):$$AI=\frac{putrescine+cadaverine+histamine}{Total\;amines}\;\cdot \;100$$
Total amines=putrescine+cadaverine+histamine+tyramine+
tryptamine+methylamine+spermidine+spermine
or the chemical index:$$Chemical\;index=\frac{putrescine+cadaverine+histamine}{1+spermidine+spermine}$$

For relationships between amine content and spoilage level for different fish species, the reader is referred to Table 2 in [23].

### 2.8. Odour

The number of citations, works, and average citations per work, regarding this attribute, are similar to ATP degradation and Biogenic amines but with a homogeneous distribution of the number of articles per year. X. Y. Huang and V. Papadopoulos are the authors with the largest number of citations regarding this attribute. Same as texture and colour, this quality attribute is usually studied in the literature together (or even correlated) with other attributes. The major aromatic compounds identified related to spoilage levels are fatty acids profiles, aldehydes, ketones, trimethyl amine (TMA), and volatile organic compounds [78]. Typically, studies focus on assessing odour following one standard sensory index (see, for example, ref. [26], for sea bass assessment). Commonly, freshness is associated with iodine shellfish and seaweed smell, and spoiled fish with muddy, putrid, faecal, pungent, smell to ammonia or ink smell in cephalopods and acidic in shrimps. Odour can be evaluated using non-invasive methods. However, using a panel of evaluators to assess this indicator is subjective and expensive. Analytical methods could be used to analyse aromatic compounds. However, in most cases, this would involve the destruction of the sample. Although, if electronic noses are properly calibrated and validated odour can be a very interesting quality indicator.

### 2.9. Colour

Despite being an important indicator for consumers, it is one of the attributes with the lowest number of citations and average citations per work. However, the interest of the scientific community seems to increase during the last 15 years. The most cited author in this field is A. Pacquit. Colour changes have been also used as an indicator for quality degradation in combination with other attributes or as a part of sensory analysis. Colour degradation kinetics are the result of changes in the pigments due to some biochemical transformations. For example, the oxidation of myoglobin and haemoglobin turns the flesh colour from red to brown. As pointed out by [6], some compounds present in fish, such as amines or ammonia, may react with the oxidised liquid causing serious browning.

The main advantage of using this feature to assess quality is that, as mentioned above, it is an important indicator for consumers. Besides, colour measurements can be obtained using non-invasive or minimally invasive techniques. However, fish skin is heterogeneous in many species. Therefore, the results provided by devices to measure colour, such as colourimeters, will vary depending on the regions of the skin being measured. A trained panel of experts could be used to globally evaluate the colour of fish skin. However, this alternative has the same disadvantages mentioned for sensory analysis and odour, i.e., it is expensive and subjective.

### 2.10. Nutrients

Fish is highly appreciated as a healthy food product [16,79,80] mainly because it is rich in nutrients such as high-quality proteins, fatty acids, and vitamins, among others. However, despite its importance, nutrients are not usually considered as a factor for determining fish freshness, except for the study of fatty acids, already considered in Section 2.1. This is the attribute with lower average citations per work, being the most cited author K. Chakraborty with 48 citations. Most of the highly cited manuscripts found in the systematic search regarding nutrients focused on different aspects [38,39,40,81], such as the characterisation of vitamin compounds; the use of fish oil; correlations between arsenic bioavailability and nutrient content; among others, not directly related to fish freshness. This is probably because the rate of degradation of most nutrients is slower than other indicators such as the QIM and when changes are noticeable, fish is already spoiled. Another disadvantage is that the assessment of nutrient content requires destructive methods.

### 2.11. Water Content/Activity

The number of citation counts and the average citations per work for this attribute are the second lowest in the list, after nutrients. This is probably because it is usually considered a stress factor and not a quality indicator itself. The most cited author is S. Cakli. Water content is related and can be described from water activity using the moisture sorption isotherm curve. Although this relationship is a non-linear function, water content increases with water activity and vice versa, and therefore both are essential quality parameters related to important textural attributes such as juiciness [12], mainly related to texture and flavour. However, as in the case of other indicators, destructive methods are used for the assessment of water content.

### 2.12. Electrical Properties

This attribute has gained the lowest attention in terms of the number of works. However, works focusing on this attribute are highly cited, with a review of different multi-sensors gathering most of the citations [24]. In fact, the average number of citations per work is the largest of all attributes considered in this review. Enzymatic and bacterial decomposition of proteins and lipids after fish death results in the formation of charged molecules which increase the electrical conductance (EC) of the muscle [6,82]. The loss of this kind of nutrients can be, therefore, correlated with the increase of EC. Autolytic spoilage is also responsible for cell membrane disruption, which allows the liquids to pour out increasing the EC [24,45].

The main advantage of this indicator is that it can be measured using minimally invasive techniques, avoiding the destruction of the sample. However, the correlation between EC increase and freshness degradation must be performed for the different fish species.

## 3. Stress Factors and Their Usual Models (Additionally, Named Secondary Models)

Food quality attributes are conditioned by the food matrix microstructure and composition as well as by several stress or environmental variables [18]. Chronologically, those factors modifying fish quality can be classified attending to the origin and slaughter conditions of the fish (pre-slaughter and slaughter) and due to handling, storage and distribution.

Fish composition and matrix microstructure depend on many factors such as the fish species, its size, age, and gender, whether it is lean or fat, fresh or saltwater fish, the fish feed, catching period (seasonal variability), geographical area or temperature of the catching waters. For example, fat, and therefore lipid oxidation, highly depends on the catching period and on whether the fish is wild or cultured [21]. Another example would be the cadaverine level which is considered a good spoilage index for wild fish, but not for aquaculture fish [23]. The slaughter procedure also affects fish quality. For example, bleeding affects lipid oxidation, preventing this oxidation in minced trout whole muscle, minced mackerel light muscle, and intact mackerel dark muscle [20]. Commonly, these factors are not modelled and, therefore, they are not considered in this review.

During handling, storage and distribution there are many external factors (stress factors) accelerating fish quality loss. Some attributes, like texture, can be affected by the just after-slaughter conditions, such as the glycolysis and *rigor mortis*, leading to gaping [63]. However, in general, outside-of-fish variables during storage and distribution are the parameters that can be manipulated to extend freshness and they are usually the focus of the mathematical models. Temperature is undoubtedly the most important and studied factor, although others such as pH [17] or CO${}_{2}$, when stored under a modified atmosphere [83], have been also considered.

There are different ways of modelling the influence of temperature. Roughly, the modelling approaches can be classified into two groups: (i) models considering a direct influence of the temperature on shelf life, and (ii) models describing the effect of temperature on the degradation of biochemical compounds or on bacterial growth, which are then related to freshness. The first type consists of pure empirical input/output relationships. These models are discussed in Section 4.1. The second type consists of mechanistic-based relationships. These models do not provide a direct input/output expression, but a function of the relationship of temperature with a kinetic parameter with a major role in any of the quality attributes or outputs.

Certainly, the most common model to describe the effect of temperature, at least when the output of the model is a product of one or more biochemical reactions, is the Arrhenius model [84]:$$K\left(T\right)=A\exp\left(\frac{-Ea}{RT}\right)\;Arrhenius\;model\;$$
where K(T) is a degradation rate that depends on the Temperature (*T*) through an exponential expression. *A* is the pre-exponential factor, Ea is the reaction activation energy, and *R* is the universal gas constant.

The influence of temperature on bacterial spoilage (output) is more complex, with many models available in the literature [60], including the Arrhenius equation. Nevertheless, the most common one is the Ratkowsky or square root model to describe the change in maximum growth rate ($\mu_{max}$) [85]:$$\mu_{max}\left(T\right)={\left[b(T-T_{min})\right]}^{2}\;Ratkowsky\;model\;$$
being $T_{min}$ the temperature at which growth is zero, and *b* the factor shaping the curvature of the function. The same functionality can be used to model the effect of temperature on the lag phase of spoilage bacteria [57].

Although the temperature is the major factor affecting fish freshness, there are other relevant stress variables such as pH (a major factor in texture [63]), water activity, salt concentration or concentration of $CO_{2}$ in packed fresh fish. The models in those cases are not so common and are of many different forms [60]. The reader is referred, for example, to the gamma concept to model the joint effect of several stress variables [86,87].

## 4. Models (Relationship between Model Inputs and Outputs)

The diversity of mathematical models for fish quality assessment emerges mainly from the diversity of the fish quality attributes previously described and from the complexity of the fish freshness concept. In the systematic review, we found 25 records where mathematical modelling of one or more attributes is included. These records [16,17,45,51,52,56,57,58,59,60,67,68,80,88,89,90,91,92,93,94,95,96,97,98,99] were revised and their main modelling information was included in the tables presented in this section. The objective of this section is to provide an overview of the most common approaches found regarding the mathematical modelling of fish quality/freshness.

In general, screened models are deterministic, lacking uncertainty analysis, and most of them are semi-empirical and described in the so-called *closed-form expression*, i.e., they are described by algebraic equations with a finite number of terms without derivatives or integrals. Fish quality usually depends on macroscopic variables that can be described using deterministic models, i.e., models without considering any random effect, thus providing the same solution for different simulations performed under the same conditions. Stochastic models, on the contrary, assume some random behaviour intrinsic in the dynamics resulting in stochastic differential equations. In that case, any realisation of the model provides a different solution. They are usually required when modelling food safety, but not for quality, where low numbers of certain variables (such as a low number of pathogenic bacteria) are decisive to assess the risk of foodborne illness [100,101]. Probabilistic models, on the other hand, lack dynamical equations or include the probabilistic part in the parameters of the dynamic. They are relevant when considering uncertainty (due to lack of information, measurement error or noise [102]) and/or variability (due to differences in the model parameters caused by, for example, changes in food matrix or spoilage bacteria strains [95]).

Attending to the type of mathematical equations, fish quality models are usually presented in their closed-form and they are based on empirical expressions used to represent certain behaviours, such as exponential growth [45,91]. When models are inspired by first principles with mechanistic or semi-mechanistic formulations, a closed-form expression may not exist, be unknown, or be too complex for practical use. In these cases, the model is directly described by differential Equations [68,96], requiring proper numerical methods for their resolution and calibration [103,104]. Models expressed in their differential form are also required when the stress variables or other model parameters (growth or degradation rates, diffusivity of a given compound, etc.) vary during storage and transportation.

When attending to the specific features of fish quality modelling, we found four different types of modelling approaches attending to their objective:**Shelf life soft sensors** are models that consider a direct input/output relationship. They consist of empirical functions, denoted by soft (from software) or virtual sensors. Typically, the input and the output are, respectively, temperature and shelf life.**Quality soft multi-sensors** are models considering a general mathematical expression that can be applied to describe more than one attribute.**Quality ad hoc models** are mechanistic-based models with equations specifically derived for one particular quality attribute.**Sensory or shelf life models** are models providing as their output a sensory score or shelf life date. However, they also require the intrinsic modelling of one or several quality indicators (such as spoilage bacterial content). To this purpose, they typically consider a quality ad hoc model. These sensors are also named *smart* when they are used not only for assessment purposes but for prediction of different degrees of fish quality as well [97].

This classification will be used to structure this section.

### 4.1. Shelf Life Soft Sensors (Input/Output)

In the literature, there are two common expressions to model the dependence of shelf life (SL) with temperature (*T*). Such expressions coincide with the ones used to model degradation rates or bacterial growth as a function of the temperature. On the one hand, sensors assume an exponential dependence of shelf life with the temperature (Shelf life decreases exponentially when increasing temperature [90]) of the form:$$SL\left(T\right)=SL_{0}\exp\left(-b\;T\right)\;Exponential\;empirical\;shelf\;life\;model\;$$
with $SL_{0}$ being the shelf life at T=0
°C and *b* a parameter that represents the degree of influence of the temperature on the shelf life. If shelf life is highly affected by temperature, parameter *b* will be large, otherwise, it will be close to zero. On the other hand, the Arrhenius empirical shelf life model [56] has been also considered in the literature:$$SL\left(T\right)=SL_{ref}\exp\left[\frac{Ea}{R}\left(\frac{1}{T_{eff}}-\frac{1}{T_{ref}}\right)\right]\;Arrhenius\;empirical\;shelf\;life\;model\;$$
where now shelf life at an effective temperature $T_{eff}$ is calculated from a reference shelf life $SL_{ref}$ at temperature $T_{ref}$. Ea is the activation energy and *R* the universal gas constant.

There are many variations of these equations such as the school-field [98], the Exponential RRS (Relative rate of spoilage) model [105] or the square-root RRS model [85,98] that were inspired by limiting the levels of spoilage microorganisms. Shelf life can also be estimated from different quality attributes, such as the level of the spoilage bacteria. However, these are more sophisticated expressions that require modelling these attributes as described in Section 4.4.

Although shelf life is an important issue, the main disadvantage of these models is that they do not provide a measurement of the current freshness state of the fish.

As shown in Table 2, only three references considering shelf life soft sensors models were found in the systematic search. Each of these references focused on one particular fish species and the authors use one or two modelling alternatives. Therefore, there is a need for works focusing on different species. A comparative study of the different modelling approaches would be also required.

### 4.2. Soft Multi-Sensors

There are general mathematical formulations that may describe major trends in growth/increase or degradation/decay of a group of attributes. The direct advantage of this approach is that the same model structure is used for different quality indicators by adjusting the parameter values to fit the experimental behaviour of each indicator. However, the mechanisms of degradation are not considered. This results in too generic expressions that are mainly based on empirical correlations. Therefore, they cannot be applied to understanding attributes with complex dynamics (such as when the property does not increase or decrease monotonically). Besides, since the mechanisms are not considered, the predictive capabilities of these models are limited to the experimental conditions used to adjust the model parameters. The results provided by this approach are less reliable than the results obtained with the *ad hoc* models presented in the following section.

Let us denote by $A_{i}$ (i=1,2,…m) a given quality attribute that depends on time (*t*), and usually on temperature (*T*). *m* is the total number of attributes considered.

The simplest multi-sensor is based on the Weighted regression coefficients model [17]. This model relates one or several outputs ($A_{i}$) with several inputs or stress variables ($S_{j}$) using a linear expression of the form:$$A_{i}=\sum_{j=1}^{w}a_{i,j}S_{j}+a_{i,w+1}\;i=1,\ldots ,m\;Weighted\;regression\;coefficients\;model\;$$

$S_{j}$ typically includes temperature, but other stress variables, such as pH, can be considered. $a_{i,j}$ are coefficients to be estimated for each attribute and each input. $a_{i,w+1}$ is the independent coefficient. Although this model is a direct input/output relationship and can be used to calculate shelf life (as in Section 4.1), the model is very general and therefore can be used to calculate many different attributes, such as the sensory freshness index [17].

Other type of multi-sensors assume that the quality attribute ($A_{i}$) behaves as a nth-order reaction as follows [51]:(4)$$\frac{dA_{i}}{dt}=K\;A_{i}^{n}\;i=1,...,m\;n\mathrm{th}\mathrm{-o}rder\;reaction\;model\;$$
where *K* is the reaction rate, which typically is considered to depend on temperature according to the Arrhenius expression. Usually, *n* is considered a natural number, although it can be any positive rational number in the so-called power law models used in other contexts. nth-order reaction models are tested in some works [51], however, the usual approach is to select *n* so that it provides the best compromise between simplicity and performance of the model (Occam’s razor principle).

Among the different expressions derived from the nth-order model, the most commonly used in the literature is the exponential model, which corresponds with a first-order reaction (n=1). Considering that the reaction rate (*K*) remains constant during the process, the expression of the closed-form is:(5)$$A_{i}=A_{i,0}\exp\left(Kt\right)\;i=1,\ldots ,m\;Exponential\;model\;or\;first\mathrm{-o}rder\;reaction\;model$$
with $A_{i,0}$ being the initial condition (initial value for attribute $A_{i}$ at t=0).

Another expression, sufficiently general to represent the attribute dynamics, is the zeroth-order reaction (n=0) model. For constant reaction rates (*K*), it results in a linear dependency of the attribute with time:(6)$$A_{i}=A_{i,0}Kt\;i=1,\ldots ,m\;Linear\;model\;or\;zeroth\mathrm{-o}rder\;reaction\;model$$

The main advantage of using the differential form, Equation (Equation 4), instead of the closed-form, Equations (Equation 5) and (Equation 6), is that it allows to consider situations where the storage or transport temperature changes.

Table 3 shows the attributes (outputs) modelled with this approach and the selected models for each case. In most cases, the growth/degradation rates depend on the temperature following the Arrhenius expression. Positive or negative values of *K* are used to represent, respectively, the increasing or decreasing evolution of the attributes. In general, the closed-form of the equations is used in these works so, as mentioned above, the temperature must be constant during storage and transport to obtain reliable results. K-value and TVB-N are the most typical indicators considered in this approach. As in the case of Shelf life soft sensors, only a few fish species were considered in these studies. More research is required to include other species.

We have also found, outside the systematic search, the use of linear models of the form of Equation (Equation 6) to describe the evolution of TAC, EC, K-value, and Sensory Analysis indicators in rainbow trout (*Oncorhynchus mykiss*) [106]. The authors in this work also compared the solutions obtained using either Arrhenius expression or Artificial Neural Networks (ANN) as a secondary model. Their results show that ANNs provide a better fit to experimental data than the Arrhenius expression, in particular for K-value and Sensory Analysis indicators.

### 4.3. Quality ad hoc Models

In the systematic search, ad hoc models were found for some quality attributes, but not for texture properties, lipid oxidation/fatty acids, non-volatile biogenic amines, other nutrients, electrical conductivity, odour, colour or water activity. Lipid oxidation, despite being the most studied quality attribute in fish, was only modelled using the generic exponential model for TBA in grass carp [91]. Non-volatile biogenic amines, in particular histamine, were commonly used to model food safety [107]. However, they were not studied for describing food quality. No models for nutrient (proteins, vitamins, etc.) degradation or water activity were found, although water activity is a factor influencing texture or bacterial growth, and included as an input in those models. Regarding colour, models are mainly proposed for processed fish [108,109,110] but no model for colour changes in fresh fish was found in the literature. Despite odour being a relevant quality indicator by itself, it is typically used in combination with other attributes, for instance, to obtain the QIM. Electronic noses could be used to obtain reliable data that could be used to calibrate and validate models describing the evolution of odour. However, in the context of fish freshness, electronic noses are used to evaluate freshness or storage time [16]. As in the case of colour, no models were found to describe the evolution of odour.

There are, however, specific models for some of the quality attributes that are explained in detail in the next subsections. These models usually consider the mechanisms of quality degradation so the results are more reliable than those obtained with the other models described in previous sections. The main disadvantage is that such mechanisms involve, in most cases, complex phenomena. Therefore, the derivation of a mathematical model in these cases is a time consuming and complex task. Models describing the evolution of spoilage bacteria are the most commonly used ad hoc models.

#### 4.3.1. Spoilage Bacteria Models Using Predictive Microbiology

Predictive microbiology is a field that focuses on modelling the behaviour of microorganisms, including spoilage bacteria, in different food matrices such as fresh fish. It is a broad area with established terms and community [60,88], and where three different models are usually considered: modelling microorganisms growth or inactivation dynamics (primary model), how these dynamics change with environmental stress or inputs variables (secondary models), and the implementation of these models in friendly software (tertiary models). The terminology of primary and secondary models can be useful to outline the different modelling approaches and will be used also in this review, even for models outside the predictive microbiology scope.

Primary models are diverse [111] and they focus on the dynamics of bacterial numbers (*N*). The most used primary models in fish spoilage bacteria are [29,112,113]:(7)$$\begin{array}{cccc} \; & N=N_{0}\exp\left(\mu_{max}t\right)\; & \; & \;Exponential\;model\\ \; & N=N_{0}+\frac{N_{max}-N_{0}}{1+\exp\left[-\mu_{max}\left(t-t_{i}\right)\right]}\; & \; & \;Modified\;Logistic\;Model\;\\ \; & N=N_{0}\exp\left(\frac{\log\left(N_{max}/N_{0}\right)}{1+\exp\left[\frac{4\mu }{\log\left(N_{max}/N_{0}\right)}\left(\lambda -t\right)+2\right]}\right)\; & \; & \;Reparametrised\;Gompertz\;Model\;\\ \; & \frac{dN}{dt}=\frac{a_{0}}{a_{0}+\left(1-a_{0}\right)\exp\left(-\mu_{max}t\right)}\mu_{max}N\left(1-\frac{N}{N_{max}}\right)\; & \; & \;Reparametrised\;Baranyi^{\prime }s\;Model\; \end{array}$$
with $N_{0}$ being the number of initial bacteria, $N_{max}$ the maximum number (in the stationary phase), $\mu_{max}$ the maximum growth rate and λ the time of the lag phase. In the modified logistic model, λ is a function of the point of inflexion ($t_{i}$):$$\lambda =t_{i}-\frac{1}{\mu_{max}}\ln\left(\frac{N_{max}+N_{max}\exp\left(\mu_{max}t_{i}\right)}{N_{max}+N_{0}\exp\left(\mu_{max}t_{i}\right)}-1\right)$$
and for simplicity in the provided equation we assume that the minimum cell number $N_{min}$ is the initial cell number $N_{0}$. The derivative form of Baranyi’s Model (closed-form solution is long and complex [114]) is presented with modifications to make lag phase zero, for $a_{0}=1$, and maximum lag (no dynamics), for $a_{0}=0$.

Works in the systematic search including modelling of spoilage bacteria are outlined in Table 4. Secondary models are only included when there is a clear description within the work. The variety of fish species considered in this case is larger than in the cases of Shelf life sensors and Soft multi-sensors. There is also a large variety of bacterial strains studied. However, most of the works found in the search consider deterministic models whereas bacterial population growth is a stochastic process. In this regard, although the mathematical structure of the model developed in [95] is deterministic, the authors estimate the variability of the model parameters using different experimental conditions and different fish samples. This variability is used to generate different combinations of parameters and each combination is used to obtain different simulation results. This approach allows us to approximate the stochastic behaviour.

#### 4.3.2. TVB-N and TMA-N Models

Volatile nitrogenous bases (TVB-N), and its major contributor TMA-N, are widely modelled quality outputs with specific modelling approaches (in addition to the general modelling [45,91]). They are easy to measure indexes that adequately correlate to fish freshness.

Firstly, Howgate [52] pointed out that the exponential model (used in general modelling approaches [45,91]) was not descriptive of TMA-N changes since TMA-N reaches a limit, instead of increasing exponentially, because they are the sub-product of TMAO. The author suggested a logistic growth of the form:$$TMA-N=\frac{TMA-N_{max}-TMA-N_{0}}{1+\exp\left(-K(t-t_{i})\right)}\;Modified\;Logistic\;Model$$
where TMA-N${}_{0}$ and TMA-N${}_{max}$ are, respectively, the initial and maximum allowed concentrations of TMA-N, *K* is the maximum growth and $t_{i}$ the point of inflexion.

On the other hand, TVB was modelled by García [97] by assuming a delay and a later production by psychrotrophic bacteria (*N*) with following equations:$$\frac{dTVB}{dt}=\left(\frac{a_{0}}{a_{0}+\left(1-a_{0}\right)\exp\left(-Kt\right)}\right)K\;N\;Exponential\;Model\;with\;delay$$
where $a_{0}$ represents the parameter determining the duration of the delay (mathematically equivalent to the expression used by [113] for lag phase in bacterial growth, $a_{0}=1$ indicates no delay), *K* is the growth rate due to psychrotrophic bacteria (*N*). Interestingly, years after this work, a simpler exponential model was used to model TVB in rohu fish stored at 0 and 5 ${}^{\circ }C$ [80], claiming that TVB formation was a primary function of microbial action and suggesting the necessity to model TVB as a function of the microbial population as already carried out in the literature [97].

#### 4.3.3. Texture Properties

As mentioned above, no predictive models were found in the systematic search for texture properties. It must be highlighted that the works [89,93] describe the models they develop as predictive. However, as mentioned in the introduction, we use the term *predictive* to indicate the ability of the model to forecast the evolution of the quality indicators. The models developed in [89,93] are built using partial least squares regression or least-squares support vector machines to assess texture indicators using nuclear magnetic resonance (NMR) or hyperspectral imaging (HSI) measurements. In other words, these models provide a non-invasive estimation of the texture properties at the NMR or HSI measurement time, but they do not predict the future evolution of such indicators.

A predictive model to describe the viscoelastic behaviour of rohu fish (*Labeo rohita*) was developed in [115], although this work was not present in the systematic search. The authors used the modified Maxwell model to relate skin hardness and compression time for iced fish:$$F\left(t\right)=C_{0}+C\exp\left(\frac{t}{t_{rel}}\right)$$
where F(t) is the force at any time, $C_{0}$ corresponds with the force at equilibrium, *C* is the decay force, and $t_{rel}$ is the relaxation time. Experimental data was used to fit coefficients $C_{0},C$ and $t_{rel}$.

Further research is required regarding the mathematical description of the evolution of texture in fresh fish.

#### 4.3.4. ATP Degradation

As mentioned in Section 2.6, ATP degradation occurs in a series of steps represented as:ATP→ADP→AMP→IMP→Ino→Hx→Xa→Uric acid

The first three steps occur relatively fast after slaughter so, when fish samples are analysed they contain low (or zero) concentrations of ATP, ADP and AMP. On the other hand, the degradation from Hx to Xa and uric acid is usually slow and when such products are formed, the fish is already spoiled. Therefore, in general, ad hoc models only consider the part of the scheme involving IMP, Ino and Hx. The $K_{I}$-value, Equation (Equation 3), can be obtained from these compounds.

Table 5 summarises the main features of the models derived in the different works of the systematic search. In particular, ref. [52,67] considered the reaction scheme:$$IMP\overset{K_{1}}{⟶}Ino\overset{K_{2}}{\underset{K_{bac}}{⟶}}Hx$$
where $K_{1}$ and $K_{2}$ are, respectively, the reaction rates for the conversion of IMP into Ino, and Ino into Hx. Bacterial conversion of Ino into Hx ($K_{bac}$) was also taken into account. In these works, first-order kinetics are considered. Arrhenius expressions were used to account for the dependency of reaction rates on the temperature. Bacterial growth was modelled using an exponential model of the form of Equation (Equation 7). In [52], the possibility of a loss of nucleotides by leaching (diffusion through muscle and skin) was also considered. Another interesting issue about this work is that the author presented and discussed the results obtained from data of forty-five different fish species. Reliable results were obtained for most of the considered species. In [68,96], the authors found, by fitting the models to experimental data, that alternative nucleotide degradation paths might occur in European hake (*Merluccius merluccius*). In particular, the direct conversion of IMP to Hx and other products should be considered. Leaching of nucleotides and the effect of bacteria, namely *Pseudomonas* spp. and *Shewanella* spp., on the conversion of IMP to Ino and Ino to Hx, were also considered. The standard square-root model [85] was used to represent the bacterial growth rates.

### 4.4. Sensory or Shelf Life Models

Quality ad hoc models are usually a tool, more than the final aim, to assess or predict shelf life or different grades of fish quality. For example, modelling of spoilage bacteria in Table 4 is commonly used to estimate shelf life by specifying a concentration of bacterial counts at which the quality is considered not sufficient. Shelf life date, for example, is estimated using spoilage bacteria in bogue fish (*Boops boops*) and gilt-head seabream (*Sparus aurata*) for numbers greater than 7 logs ($N>10^{7}$ CFU/g) [56,57] or even $N>10^{8.5}$ CFU/g for *Pseudomonas psychrofila* in tropical shrimp [59].

There are models in the literature that are used to estimate different grades of freshness, not only shelf life or a given quality indicator. These models are summarised in Table 6. As shown in the Table, works following this approach and the number of species considered are scarce. The main challenge is to find a mathematical relationship between the outputs of the ad hoc models and different quality levels. The first attempt consisted of dividing the QIM into three indexes ($QIM_{S}$ for skin, $QIM_{G}$ for gills, and $QIM_{F}$ for flesh), and finding their relationship with non-producers ($N_{w}$) and producers ($N_{b}$) of sulphide, present in those specific parts for the fish [92]. However, for an estimation of a final QIM index, not only models of spoilage bacteria but also of TVB-N are required [97]. In this work, a simple ANN was developed to obtain the relationship between QIM and the model variables

(TVB-N and bacterial count). A logistic model was used to describe the bacterial evolution. Other works focused on using models of spoilage bacteria to find ranges of standard sensory methods. Such methods considered fewer freshness grades than the QIM. That is the case in [16], where cod freshness, in terms of a three-level standard (SC/T 3108-1986), was correlated with the TVC value. The work by [95] used a nonlinear function of *Pseudomonas* and *Shewanella* counts to determine a four-level QSM value in European hake. This work is the only one that considers a secondary model, and therefore, it is the only one that provides the final relationship between the effect of temperature changes in the four levels of quality in QSM.

## 5. Modelling Challenges and New Directions

In this review, we have described, classified and established connections for the different types of mathematical models to describe and predict fish quality. We have mainly focused on those works within the systematic search described in Appendix A, although other relevant contributions, not included in such search, were considered in this review. First of all, we should stress that the literature is larger, particularly in the case of spoilage bacteria using predictive microbiology. However, there are many comprehensive reviews regarding predictive microbiology [60,116,117], and the focus of this review was, instead, on providing connections between models that describe different fish quality attributes, and on identifying those topics that require more research attention. For example, there are still relevant quality attributes for which it was not possible to find mathematical models. The most illustrative example is lipid oxidation, the most studied attribute in the literature from the experimental point of view. Other attributes lacking modelling approaches are non-volatile biogenic amines, nutrients, odour and water activity (although this is considered as an input in several models in the literature). Other attributes, such as colour, have been modelled only for processed fish.

We need to stress that, in addition to the lack of modelling for certain quality attributes, there are major limitations in some of the models we found. Generic models, named in this review software sensors or multi-sensors, are particularly advantageous to compare results from different studies but, in this comparison, it is clearly observed that there are many inconsistencies between works. For example, both linear and exponential functions have been used to model the same attribute (see Table 3). However, whereas linear functions may approximate a short time window of an exponential model, differences between both approaches are considerable for wide prediction windows. There is a need to compare those modelling structures and detect which ones are more appropriate for the different quality outputs.

*Ad-hoc* predictive microbiology models, that have been studied in detail, still present limitations, mainly due to the uncertainty of the estimated parameters and the initial bacterial fish load (or model initial conditions). Microbial models require a known starting state, but measuring bacterial load takes time and involves the destruction of the sample. Some partial solutions have been considered such as (1) using the worst-case scenario [116], (2) estimating the initial conditions variability [95], or (3) estimating the numbers using indirect measurements of other variables (such as conductance measurements) that are non-invasive and fast to obtain [118]. However, taking into account that bacteria grow exponentially between lag and stationary phase, model prediction is highly affected (very sensitive) by its initial conditions. More research on this topic is required to find confidence models of spoilage bacteria.

Another challenge, still only partially addressed in the literature, is the derivation of expressions that allow the inference of sensory attributes or shelf life from the growth of spoilage bacteria. In this regard, some works derived models describing two or several attributes, for instance, shelf life and growth of *Pseudomonas*, but such attributes were only connected through the stress variables, typically temperature (see examples in Table 4). Ideally, the model should provide a final quality index, as a function of different quality attributes that depend on the stress variables, as the examples provided in Table 6.

In addition, only a few works validate the predictive capability of the models proposed, i.e., the ability of the model to describe data outside of the set used for model development and parameter estimation. Most of them use constant temperature or temperature oscillating at a high frequency, as compared with the model dynamics time-scale (using such oscillating temperature would be equivalent to using a mean constant temperature). For example, in the work by [56] two non-isothermal profiles are used for the validation. In one profile, temperature oscillates at high frequency and the change in the output signal was smaller than the experimental error of the measurement. On the other hand, the model was validated using a temperature profile with wider oscillations, that provided a change in data trend and model dynamics. Only a few works consider dynamic temperature profiles, computed using an optimal experimental design, to reduce the uncertainty of the predictive model [95].

To advance in this area, reproducibility of published works is a key aspect, particularly for ad hoc models, usually more complex and with many different mathematical structures. Currently, comparison among works is extremely difficult, not only because of the variability between fish species and conditions before and after the capture (for example in food structure [119]) but also because the proposed measure of fish quality is sometimes specifically developed for the study [118,120]. We think that for the advance of modelling of fresh fish quality, research should be focused on reproducing and predicting established sensory indexes, such as QIM [95,97,121], allowing comparison between approaches.

Finally, the community should invest in better exploitation of the available models and towards their integration into software systems for online quality prediction, as is already the case in food safety [122], or even for optimisation-based determination of the best conditions to maximise shelf life in different processes. In this regard, an ambitious objective would be the derivation of a digital twin [123] for fresh fish degradation. Let us here use an illustrative example of the potential of these models in a study developed in our group, namely [124]. In this work, existing models were used to find the best active package configuration (including the type of packaging and concentration of antimicrobial) that maximises food quality while ensuring food safety. In addition, the model was used to predict, at any moment, the expected food quality for the expected stress variables along the food chain.

## Figures and Tables

**Figure 1 foods-11-02312-f001:**
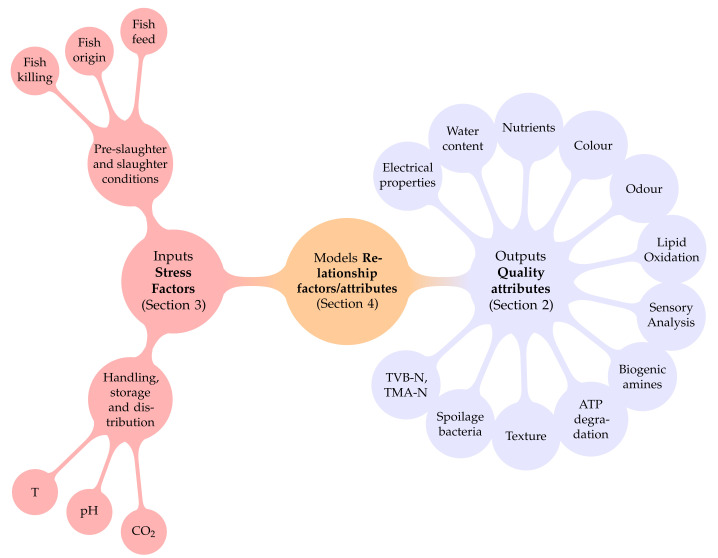
Fish quality models are described attending to their Quality attributes (Section 2), Stress factors (Section 3) and models, i.e., mathematical relationships between attributes and stress factors (Section 4).

**Figure 2 foods-11-02312-f002:**
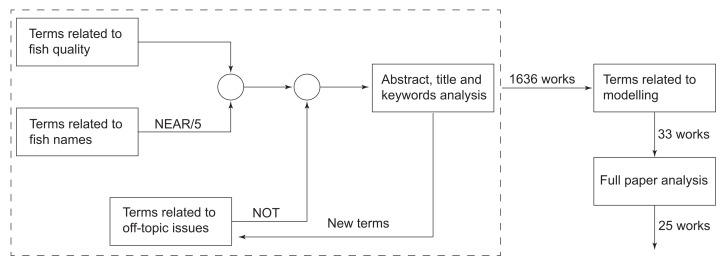
General scheme of the procedure followed in this manuscript to perform the systematic literature search.

**Table 1 foods-11-02312-t001:** Most employed quality attributes in the literature. The number of total citations per year is used to obtain the most cited articles. The terminology used for each attribute was: Lipid oxidation (fatty acid*, lipid oxidation, TBA, TBARS, thiobarbituric), Sensory analysis (QIM, QSM, sensory analysis, sensory evaluation, sensory method, TVB-N/TMA-N (TVB-N/TMA-N), Spoilage bacteria (SSO, spoilage bacteria, spoilage microorganism*), Texture properties (texture, hardness, firmness), Biogenic amines (biogenic amine*), Odour (odour, odor), Colour (colour, color, chromatism), Nutrients (nutrient*, vitamin), Water content/activity (water content, water activity) and Electrical properties (electrical properties, conductance, conductivity).

Quality Attribute	Citation Counts	No. Works	Avg. Citations per Work	Most Cited Works
Lipid oxidation	10,875	474	22.9	Herrero [12], Richards and Hultin [20], Grigorakis et al. [21]
Sensory analysis	5295	209	25.3	Ólafsdóttir et al. [22], Al Bulushi et al. [23], Olafsdottir et al. [24]
TVB-N, TMA-N	4386	185	23.7	Pacquit et al. [25], Papadopoulos et al. [26], Ruiz-Capillas and Moral [27]
Spoilage bacteria	3876	155	25.0	Al Bulushi et al. [23], Gram et al. [28], Dalgaard [29]
Texture	3587	176	20.4	Herrero [12], Olafsdottir et al. [24], Alasalvar et al. [30]
ATP degradation	3509	124	28.3	Ólafsdóttir et al. [22], Veciana-Nogués et al. [31], Jones et al. [32]
Biogenic amines	3282	112	29.3	Al Bulushi et al. [23], Veciana-Nogués et al. [31], Kim et al. [33]
Odour	3066	119	25.8	Papadopoulos et al. [26], Ramanathan and Das [34], Kawai [35]
Colour	2830	149	19.0	Pacquit et al. [25], Kuswandi et al. [36], Huang et al. [37]
Nutrients	704	62	11.3	Chakraborty and Raj [38], Moreda-Piñeiro et al. [39], Palaniappan and Vijayasundaram [40]
Water content/activity	573	32	17.9	Cakli et al. [41], Morzel et al. [42], Raju et al. [43]
Electrical properties	381	11	34.6	Olafsdottir et al. [24], Vaz-Pires et al. [44], Yao et al. [45]

**Table 2 foods-11-02312-t002:** Summary of the shelf life soft sensors models found in the literature search.

Output	Matrix	Model	References
Shelf life	Bogue	SL(T) Arrhenius emp.	Taoukis et al. [56]
Shelf life	European sea bass	SL(T) Exponential emp.	Limbo et al. [90]
Shelf life	Large yellow croaker	SL(T) Exponential emp. & school-field	Quanyou et al. [98]

**Table 3 foods-11-02312-t003:** Virtual multi-sensors (same model structure for modelling different attributes). In the table TVB-N = total volatile base nitrogen, TAC = total aerobic counts, EC = electrical conductivity, GSI = global stability, SL = Shelf life, SFI = Sensory Freshness Index index, TM = Torrymeter reading, IT = Internal Temperature, ST = Superficial Temperature.

Output	Matrix	Secondary Model	Primary Model	References
TVB-N, TAC, K-value	Grass carp	K(T) Arrhenius	Exponential model	Zhang et al. [91]
TVB-N, TAC, K-value, EC	Crucian carp	K(T) Arrhenius	Exponential model	Yao et al. [45]
GSI (Sensory Score, TAC, TVB-N, K-value)	Bighead carp	K(T) Arrhenius	Linear model	Hong et al. [51]
GSI (sensory score, K-value, TAC and TVB-N), EC	Crucian carp	K(T) Arrhenius	Linear model	Zhu et al. [94]
SL, SFI	Gilt-head seabream	$A^{i}$(pH, TM, IT, ST, TVB-N)	Weighted regression coefficients.	Calanche et al. [17]

**Table 4 foods-11-02312-t004:** Works in the systematic search, including modelling of spoilage bacteria. The following acronyms are used: TVA for total viable counts, TMAB for total mesophilic aerobic bacteria, TPAB for total psychrophilic aerobic bacteria and LAB for lactic bacteria.

Output	Matrix	Secondary Model	Primary Model	References
*Pseudomonas* & *Shewanella*	Bogue	$\mu_{max}\left(T\right)$ Arrhenius & Ratkowsky	Baranyi’s model	Taoukis et al. [56]
*Pseudomonas* & *Shewanella*	Gilt-head seabream	$\mu_{max}\left(T\right)$&λ(T) Arrhenius & Ratkowsky	Mod. logistic model	Koutsoumanis and Nychas [57]
Sulphide producers & non-producers	Gilt-head seabream	$\mu_{max}\left(T\right)$ (not clearly defined)	Baranyi’s model	Giuffrida et al. [92]
*Pseudomonas* & *Carnobacterium*	Tropical shrimp	$\mu_{max}\left(T\right)$ Arrhenius & Ratkowsky	Baranyi’s model Rep. Gompertz Model	Dabadé et al. [59]
*Pseudomonas* & *Shewanella*	Hake	$\mu_{max}\left(T\right)$ Ratkowsky	Baranyi’s model	García et al. [95]
TVC	Grass carp	–	Rep. Gompertz Model	Ying et al. [16]
Psychrotrophic counts	Cod	–	Baranyi’s Model	García et al. [97]
*Pseudomonas*, *Enterobacteriaceae*, TMAB, TPAB & LAB	Rainbox trout	$\mu_{max}\left(T\right)$ Ratkowsky	Mod. Logistic Model	Genç and Diler [99]
*Pseudomonas*	Gilt-head seabream	–	Mod. Logistic Model	Correia Peres Costa et al. [58]
Biomass	Rohu fish	–	Mod. Logistic Model Gompertz Model	Prabhakar et al. [80]

**Table 5 foods-11-02312-t005:** *Ad-hoc* models found in the systematic search to describe the degradation of IMP, Ino and Hx. The $K_{I}$-value is obtained from the concentration of these components. All these models consider a cascade of first-order reactions.

Output	Matrix	Secondary Model	Primary Model	References
IMP, Ino, Hx	Rainbow trout	$K_{i}\left(T\right)$ Arrhenius	Exponential model, Bacterial catalysis	Howgate [67]
IMP, Ino, Hx	Forty-five species	$K_{i}\left(T\right)$ Arrhenius	Exponential model, Bacterial catalysis, leaching	Howgate [52]
IMP, Ino, Hx	Hake	$K_{i}\left(T\right)$ Arrhenius	First-order reaction model	Vilas et al. [96]
IMP, Ino, Hx	Hake	$K_{i}\left(T\right)$ Arrhenius	First-order reaction model, Bacterial catalysis, leaching	Vilas et al. [68]

**Table 6 foods-11-02312-t006:** Modelling of sensory scores using ad hoc models. QIM stands for Quality index specific for gilt-head seabream [65] or cod [50] method. S,G and F for Skin, Gills, Flesh, $N_{w},N_{b},N_{p},N_{s},N_{psy}$ for sulphide and non-sulphide produces, *Pseudomonas*, *Shewanella* and psychrotrophic counts.

Output	Matrix	Secondary Model	Primary Model	References
$QIM_{S},QIM_{G},QIM_{F}$ (15 levels)	Gilt-head seabream	Not clearly defined	$QIM(N_{w},N_{b})$	Giuffrida et al. [92]
Council Regulation(EC) No 2406/96 (1996) Standard method (4 levels)	Hake	$\mu_{max}\left(T\right)$ Ratkowsky	$SM(N_{p},N_{s})$	García et al. [95]
SC/T 3108-1986 Standard method (3 levels)	Cod	–	$SM(N_{TVC})$	Ying et al. [16]
QIM (23 levels)	Cod	–	$QIM(TVBN,N_{psy})$	García et al. [97]

## Data Availability

All bibliometric analysis is shared in the repository [19], with link doi:10.5281/zenodo.6414360.

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
