# Peer review of "Assessment and Prediction of Fish Freshness Using Mathematical Modelling: A Review"

_foods, 2022, doi:10.3390/foods11152312_

Round 1

Reviewer 1 Report

Revision: Assessment and Prediction of Fish Freshness using Mathematical Modeling: A Review

 1.               General comment:

This review provides an inclusive vision of the mathematical models used to describe and predict the evolution of the main quality indicators in fresh fish and shellfish during storage.

The reviewer considers that this work will be very useful for rethinking future trials in this area. It should, however, be said that some words, without any information, as is the case of ...percepton model with logistic activation (Line 529), make reading difficult. So, for example, in this case, it could be indicated which is the difference between logistic regression and perceptron with logistic activation.

 2.               Some comments

Title: Modeling is American English. In British English is Modelling

Line 9: … describe and forecast the degradation of the key quality indicators…

Suggestion: …describe and predict the changes of the key quality indicators…

 Line 20: Fresh fishery products (FFP) are very perishable foods stored, transported and

distributed in boxes of high-density poly-ethylene filled with ice.

Suggestion. Fresh fish and shellfish are highly perishable products due to their biological composition. Under normal handling and chilled or refrigerated storage conditions, their shelf life is limited by enzymatic, chemical and microbiological spoilage.

Line23: From fish catch to final sell to the consumer

Suggestion: From fish capture to consumer consumption

 Line 74: On the other extreme, quality attributes, such as freshness, are defined by the combination of nutritional and organoleptic properties which deteriorate with time.

The authors do not consider here the chemical and microbiological due to any reason?

 Line 82: including terms related with fish freshness and rejecting terms associated with processed fish or other topics which are out of the scope of this work. The list of terms used in the systematic search is presented in Appendix A.

 Line 91: …separation of five words (NEAR/5) between…

Why NEAR/5?

 Line 284: Fish composition and matrix micro-structure depend on many factors such as the fish type or species (lean/fat fish, fresh/saltwater fish), fish feed, catching period (seasonal variability) or temperature of the catching waters.

Suggestion: Fish composition and matrix micro-structure depend on many factors such as the fish species (size, age, gender, lean/fat fish, fresh/saltwater fish), fish feed, catching period (seasonal variability), geographical area or temperature of the catching waters.

Line 519

Please check if is boque fish or bogue fish. In the original paper it is boque fish, but this is not the common name of Boops boops

Line 520 and Table 6

Usually, the common name used is gilthead seabream (Sparus aurata). Please check.

Reviewer 2 Report

The manuscript deals with a review of the assessment and prediction of fish freshness using mathematical modeling.

 This review is interesting. Nevertheless, the authors must rearrange the manuscript and discuss in more detail the advantages and disadvantages of the different systems, rather than showing, in some cases, a compilation of several different studies. Moreover, the text should be balanced with the presented figures and tables.

 The English language must be revised.

Please format all scientific names in italic.

Around 84 references have more than 5 years. Please update your list of references.

Round 2

Reviewer 2 Report

The manuscript was improved.